# Diving in Nose First: The Influence of Unfamiliar Search Scale and Environmental Context on the Search Performance of Volunteer Conservation Detection Dog–Handler Teams

**DOI:** 10.3390/ani11041177

**Published:** 2021-04-20

**Authors:** Nicholas J. Rutter, Tiffani J. Howell, Arthur A. Stukas, Jack H. Pascoe, Pauleen C. Bennett

**Affiliations:** 1Anthrozoology Research Group, School of Psychology and Public Health, La Trobe University, Bendigo, VIC 3552, Australia; T.Howell@latrobe.edu.au (T.J.H.); Pauleen.Bennett@latrobe.edu.au (P.C.B.); 2School of Psychology and Public Health, La Trobe University, Bundoora, VIC 3083, Australia; A.Stukas@latrobe.edu.au; 3Conservation Ecology Centre Cape Otway, Cape Otway, VIC 3233, Australia; jack@conservationecologycentre.org

**Keywords:** detection dog, detection dog training, search performance, search sensitivity, search context, search strategy, performance generalization, environmental influence, conservation detection dog, volunteer

## Abstract

**Simple Summary:**

Conservation detection dogs (CDDs) are trained to locate biological material from plants and animals of interest to conservation efforts and are often more effective and economical than other detection methods. However, the financial costs of developing and appropriately caring for CDDs can make them inaccessible for smaller conservation organizations. Training skilled volunteers to work with suitable pet dogs may help increase accessibility. We sought to further develop the skills of 13 volunteer dog–handler teams that were trained in a previous study to detect myrrh essential oil in controlled laboratory conditions. We recorded the proportion of targets found, false alerts made and search duration of the dog–handler team group through progressive training stages outdoors that increased in size and environmental complexity. First, teams searched various-sized areas before and after 12 weeks of search training on a sports-field. Next, teams searched various-sized areas before and after seven weeks of training in bushland. Overall, teams found approximately 20% fewer targets in each unfamiliar context, compared to performance in familiar contexts. However, teams typically found 10–20% more targets after a period of training compared to baseline searches. Search performance varied between teams, yet six teams found at least 78% of targets after training in bushland. Our results help to validate our stepped approach to training and highlight the need to train volunteer CDD teams to work in various-sized areas and environments.

**Abstract:**

Conservation detection dogs (CDDs) are trained to locate biological material from plants and animals of interest to conservation efforts and are often more effective and economical than other detection methods. However, the financial costs of developing and appropriately caring for CDDs can nonetheless prohibit their use, particularly by smaller conservation organizations. Training skilled volunteers to work with suitable pet dogs may help address this constraint. We sought to further develop the skills of 13 volunteer dog–handler teams that were trained in a previous study to detect myrrh essential oil in controlled laboratory conditions. We assessed search sensitivity, search effort, search precision and false-alert instances through progressive training stages increasing in size and environmental complexity. First, teams searched various-sized areas before and after 12 weeks of search training on a sports-field. Next, teams searched various-sized areas before and after seven weeks of training in bushland. Overall, search sensitivity decreased by approximately 20% in each unfamiliar context, compared to performance in familiar contexts. However, sensitivity typically improved from baseline performance by 10–20% after a period of training. Six teams found at least 78% of targets after training in bushland, yet sensitivity ranged from 29% to 86% between teams. We maintain that the foundational skills developed previously were necessary to prepare volunteer teams for field surveys involving conservation related targets. However, our results highlight the need to also train volunteer CDD teams in search scale and environmental contexts similar to their intended working conditions.

## 1. Introduction

Conservation detection dogs (CDDs) are trained to locate biological material from plants and animals relevant to conservation efforts [1]. The olfactory sensitivity of dogs and their desire to work cooperatively with people has enabled their use in diverse conservation applications, detecting a variety of targets from floating whale scats [2] to regurgitated owl pellets [3]. Compared to existing survey methods, CDDs can be more effective and more economical in determining the presence/absence and abundance of plants and wildlife in a relatively non-invasive way [4,5,6,7,8]. As species extinction rates increase internationally [9], so do the costs of monitoring cryptic and endangered species. Therefore, developing effective and economical survey methods is important.

Models of procuring, training and utilizing CDDs are varied [10,11,12] and poorly documented. However, we are aware that procuring and training working dogs, while also meeting their housing, nutrition, veterinary and welfare requirements, imposes considerable time and financial costs. These costs are prohibitive for many organizations that could benefit from CDD use. Furthermore, working dogs are often kenneled when not working, which can lead to sub-optimal welfare [13,14] and compromised working performance. Development of a model for training and deploying skilled volunteers and their pet dogs in some CDD roles could help address these financial and welfare issues. Participants in a volunteer-based model would be responsible for their dog’s care, nutrition and veterinary costs, and provide an enriching home environment throughout and beyond the dog’s working life [11].

Browne et al. [10] demonstrated that volunteer dog–handler teams could be trained to identify reptile odors in a “match to sample” task in which odor locations were also visually salient. Rutter et al. [12] subsequently showed that, following appropriate training, volunteers and their pet dogs could detect an odor in controlled indoor conditions without salient visual clues. This is important information but, of course, working indoors in controlled conditions is very different from working outdoors, where distractions are numerous, environmental conditions fluctuate widely and the potential risks to the dogs, their handlers or other animals is much greater.

When training and working, detection dogs are often required to use skills learnt in one context and apply them in another context, which is termed generalizing [15]. This term can refer to either context generalization; where dogs are required to apply search skills in different contexts (e.g., different environments, search sizes or other variables) [16,17,18], or olfactory generalization; where dogs are required to categorize stimuli (i.e., odors) that are perceptually similar as likely to share an associated outcome (i.e., reward) [19,20,21,22]. Training of professional detection dogs, such as those who work in military, customs and police services, typically includes context and olfactory generalization, in which dogs and their handlers are exposed to a range of environments and odors before being deployed. Context generalization is likely to be of particular importance to CDDs.

Little is known about how volunteers trained in a laboratory might perform in larger, field-based search environments where weather and sensory distractions are present and where searches may require more handler input than in controlled room searches. Also unknown is the amount of extra training required to transition volunteers from the laboratory to the field. Environmental factors are particularly relevant to CDD work, as vegetation, topography and weather, can impact odor movement, influencing odor detectability and, consequently, search performance [16,18,23,24]. Search scale may also be relevant, particularly when searching areas larger than dog–handler teams are accustomed to.

Although search performance has been related to various environmental factors [16,18,25], very little peer reviewed research has explored how search performance developed in one context can generalize to another [16,17]. This is valuable as CDD–handler teams familiar with working in one environmental type may be required to search a different context for the same or different target odors, yet search performance in one context may not be the same in another [16,17,18]. Furthermore, while some research has explored the influence of repetitive search tasks on search performance [26], a robust analysis of whether provision of additional training in situ improves search performance is lacking from the literature. When working outdoors in natural environments, the skills of the handler become critical and often involve developing and implementing a search strategy regarding how they and their dog will search an area. This typically takes into account factors such as wind, weather conditions, topography, vegetation and survey objectives and is often adjusted in response to changing environmental conditions, such as wind direction. There is currently no agreed upon best practice search strategy for any given area [27,28] yet having a strategy in place is important in helping handlers confidently determine which areas have or have not been searched.

In a previous study [12], volunteers were trained with their pet dogs to search for an odor in a highly controlled indoor context. In this study, we extended this training to working outdoors, using the same teams of dogs and handlers in two different experiments. The overall aim in this study was to explore how well volunteer dog–handler team search performance acquired in one context can generalize to searching in unfamiliar-sized areas and in unfamiliar search environments. In the first experiment of this study, we sought to understand how well the volunteer dog–handler teams could conduct an outdoors search in what we called a ‘simple field condition’; a well-maintained sports field. In the second, we assessed their skills in a ‘complex field condition’; a box-ironbark woodland preserve. In both conditions, we evaluated their performance before and after several weeks of training and in plots that varied in size.

## 2. Experiment 1: Simple Field Conditions (SFC)

### 2.1. Materials and Methods

#### 2.1.1. Participants

Fourteen teams of volunteer handlers and their pet dogs were recruited from a previous study where they had learnt to search for a target odor, myrrh, in a controlled indoor environment [12]. Relevant demographic details are presented in Table 1.

#### 2.1.2. Materials

Myrrh essential oil (Leonardi Laboratories^®^, West Ryde, NSW, Australia) was used as the target odor and presented in two ways. 1. Approximately 0.025 mL–0.05 mL of myrrh oil was absorbed onto the fibrous end of a cotton tip (Black & Gold^®^, Macquarie Park, NSW, Australia and Swisspers^®^, Auckland, New Zealand) presented in one of 10 Polyvinyl Chloride (PVC) ‘scent pots’ [12] on a scent board (Figure 1), or placed directly on the ground. 2. Approximately 0.05 mL–0.20 mL of oil was absorbed onto paper towel packed into open-ended steel tubing (approximately 20 mm × 80 mm). Unscented control pipes and tips were also used in training and assessments to discourage dogs from inadvertently learning to detect non-target material. Scented pipes were marked with an “m” with a permanent marker while control pipes bore a “~”. Pipes were periodically cleaned with disinfectant wipes (Strike^®^, Abbotsford, VIC, Australia). Scented and control pipes were stored in separate, airtight containers. Disposable gloves were used when handling pipes. Disposable gloves and tweezers were used to handle tips.

#### 2.1.3. Procedure

Participants trained weekly in groups of four or five, with sessions designed to train teams, already experienced in searching controlled indoor environments, to search in a well-maintained grassed sports field, subsequently referred to as a Simple Field Condition (SFC). Three different-sized areas or ‘plots’ were used in experiment one; small 10 m × 10 m (100 m^2^) areas, medium 25 m × 25 m (625 m^2^) areas and large 50 m × 100 m (5000 m^2^) areas.

The 12 week training phase consisted of ten, two-hour classes held weekly and two, two-day workshops held at the beginning and middle of the training phase in which teams further developed their searching skills in more detailed, four-hour classes. Training consisted of exclusively reward-based methods, whereby each dog was given an opportunity by its handler to search for the target scent and was rewarded for finding it [12,29,30,31]. All training sessions began with each team being briefly permitted to explore the training area to satisfy interest in potentially distracting odors and help focus dogs’ attention on training. Training was delivered by professional CDD trainers/handlers from two different organizations. The SFC sites contained many naturally occurring scents, including kangaroo scats, which served as non-target odors that the dogs were required to ignore.

#### 2.1.4. Familiarization

Each team completed an initial familiarization session in the new environment to help ensure dogs would be searching, rather than simply walking around without understanding the activity at hand, as this would limit the validity of a test of context generalization. In addition to this fundamental understanding, these basic activities were intended to give the dogs (and handlers) sufficient confidence to search the new environment without causing frustration or distress, particularly around dogs not understanding what their owner is asking them to do. This was important as experimenters did not want to cause negative experiences that may impact dogs’ welfare or cause them to lose interest in searching. These activities all took place within approximately 3 m × 4 m (12 m^2^) and began with each team performing a simple search with a scented target pot and two non-target pots, all placed on the scent board, which was placed on the ground in a corner of the search area. Dogs were rewarded for investigating the target pot, even if they did not display their full ‘alert behavior’ (i.e., dropping and repeatedly nosing the target pot or freezing over the target pot). Two to three non-target pots were gradually added over three to five successful trials until the teams were competently conducting full, 10-pot board searches and displaying full alert behaviors. Dogs were then encouraged to search independently of the scent board, using the same methodology described by Rutter et al. [12]. Initially in this stage, the target pot was placed just off the end of an otherwise full scent board with no odor cue. After two to four successful repetitions, the target pot was moved 10 cm further away, to a maximum distance of approximately 50 cm.

Next, the target pot was repositioned close to the board and the mesh cap removed. This exposed dogs to the myrrh-infused cotton tip, which was taped to the floor of the pot to avoid accidental ingestion. After one or two trials, the tip was detached from the pot, which was removed, and the tip was positioned in place of the pot, next to the board. Familiarization then followed the same procedure as described above, except the tip remained in its known position while the board was moved increasingly further away until it was removed entirely over approximately three to five trials. Most teams could progress from searching indoors to searching off the board within the first two familiarization rounds (approximately 30 min of training per team representing approximately 10–20 reinforcement instances in total). Once each dog was competently performing this familiarization task, teams completed a small 10 m × 10 m (100 m^2^) area search, during which the trainers provided basic instructions to the handlers on how to search outdoors. Each team then completed the pre-training (i.e., baseline) assessment, as described below. No further familiarization tasks were conducted in experiment one.

#### 2.1.5. Training Methodology

In subsequent training sessions, handlers were trained to begin searches downwind from the target odor. Initially, searches began 3–5 m downwind from the scented cotton tip. Hand gestures and verbal cues were used to send the dog in the desired direction and handlers quickly rewarded dogs for investigating the tip regardless of alert behavior. Once trainers determined teams were competent (typically after five presentations), they progressed to the next activity, whereby teams commenced searching outside of a scent cone [32] and progressed into it.

This activity was scaled up over successive trials with a new transect being added approximately 10–20 m downwind from the original transect (Figure 2). 

Handlers determined a comfortable pace to move with their dog to search areas up to medium size (25 m × 25 m; 625 m^2^), while gradually increasing the time between when their dogs alerted and when they received a reward. False alerts were not rewarded with food, but handlers often verbally encouraged their dog and gave them the search cue again.

While there are numerous approaches to designing search strategies with CDDs [28], dog–handler teams in this study were then taught to begin searching upwind of the search plot, i.e., along the edge of the plot that the wind is coming from. Teams begin inside the plot and by progressing through parallel transects in this manner, dogs can sample a new ‘corridor’ (area between transect lines) of odors with each new transect. Compared to commencing downwind of the plot, beginning upwind means dogs are less likely to detect and follow odors in distant ‘corridors’ at the expense of searching the plot more systematically, and handlers can more confidently determine which areas within the plot have or have not been searched (Figure 3). Scented targets were collected after they are found.

As multiple teams were trained in one location due to constraints on the available space, targets were frequently moved around the area, which likely left some level of residual odor. Alerts on suspected residual odor were not rewarded which likely resulted in dogs learning that only alerting to a minimum threshold of odor (i.e., whole tip or pipe) resulted in a food/play reward. The dogs appeared to learn this new ‘rule’ within one to five instances of alerting to residual odor and subsequent instances were uncommon.

#### 2.1.6. Search Assessments

Dog–handler performance was assessed twice in Experiment 1. The first assessment was conducted on Day 1 of training and followed immediately after the familiarization process described above, with teams completing a medium SFC baseline search assessment in 25 m × 25 m (625 m^2^). In this initial search, teams were given five minutes and instructed to search the area as best they could (one target was present). After twelve weeks of training in the small and medium-sized SFC, the performance of each team was measured in four consecutive medium SFC searches (two groups of two searches separated by a break of approximately 30–45 min) to determine the effect of training on performance. In these trials, teams were given 10 min to search the area as best they could (two targets were present). Although increasing the search time made pre- and post-familiarization comparisons difficult, this was deemed necessary due to observations made during the training process, that more time was necessary to ensure all teams had adequate time to effectively search the whole area. After a break of approximately one hour, we then assessed the influence of unfamiliar search scale on performance after the 12 week training period, using a large area search of 50 m × 100 m (5000 m^2^); eight times larger than teams were familiar with (four targets were present). This search had a maximum time limit of 20 min.

Throughout all assessments, a total of eight medium plots were available and a maximum of two searches were conducted per plot each day in order to reduce the potential for dogs to be cued by the tracks of teams from previous searches. Due to space restrictions, all large searches were conducted in the same area, but different target locations were used to minimize cueing the dogs to locations with lots of dog and handler odors. Teams did not observe other teams’ searches.

In all searches, target odor locations were predetermined by dividing search plots into 5 m × 5 m grids, from which target position coordinates were pseudo-randomly generated, such that the same location was not used for the same dog twice. This prevented teams locating targets by memory instead of olfaction. Targets were positioned no more than 20 min prior to searches commencing. Targets (i.e., either myrrh-infused cotton tips or myrrh-infused paper towel packed into steel pipes) were placed directly on the ground, in between blades of grass and were not visually obvious to humans. This was very effective; researchers had difficulty recovering unfound targets on several occasions. Two to four un-scented control cotton tips and steel pipes were included in all searches and were also placed according to grid references. False alerts to these and other objects naturally present in the environment were combined and reported as an overall number of false alerts during target-present and target-absent (control) searches. This enabled experimenters to monitor the overall number of false alerts during target-present and target-absent (control) searches. Target-absent searches were interspersed between target-present searches of the same size and context. Teams were blind to the number and position of targets and un-scented control objects in all searches. While experimenters could not entirely avoid laying human scent tracks when placing target odors in search areas, experimenters ensured they did not walk directly to target locations to put them down but rather approached in indirect zig-zag or loop patterns to avoid cueing dogs to target locations. Experimenters also inevitably laid scent tracks when laying out traffic cones to designate search areas at the beginning of each assessment day, during each assessment and when retrieving unfound targets at the end of an assessment. This led to search areas quickly becoming saturated with numerous human tracks which reduced the likelihood of dogs being cued to target locations through experimenter tracks. Dogs in this study were not observed to follow tracks to locate targets. New tips were used for each search and steel pipes were cleaned at the end of each assessment day.

Searches were conducted with dogs both on and off long leads (approximately 10 m in length) depending on owner preference and training progress. Handlers initiated a search by giving their dog a “search” cue and communicated to experimenters that an odor had been located by declaring “found” when they believed their dog was alerting to a target. Experimenters confirmed whether alerts were true positives by replying “correct” or false positives by replying “incorrect”. Handlers rewarded dogs for true positives and ignored false positives, and the search continued. Experimenters collected scented targets as teams found them. Dogs were occasionally observed indicating on previous target locations during training and assessments. During assessments, these instances were recorded as false positives. Trials concluded after a set time limit had elapsed or when the handler declared that no targets remained in the search area. Odor-absent searches were scored as correct when handlers either did not declare an odor to be found within the time limit (i.e., no alert) or declared the area to contain no odor within the time limit. Search effort, the duration of each search [33], was recorded and descriptive statistics are reported in Table 2. Experimenters observed all searches, either from the side of the search plot or from following approximately 3 m behind the handler, mimicking their pace and movements to avoid cueing. Dogs were accustomed to the experimenters’ presence while searching and had learnt during training sessions that soliciting interactions during searches would be ignored.

#### 2.1.7. Data Analysis

Search performance was measured by sensitivity (i.e., proportion of targets found relative to the total number of targets available) [33] and false-alert instances. We also report search precision (i.e., the proportion of alerts directed towards a true target) [33]. Kolmogorov-Smirnov tests revealed that search sensitivity and false-alert instances were not normally distributed due to predominately correct responses. Therefore, non-parametric statistics were used for analyses involving these variables [34]. Wilcoxon Signed Rank Tests (WSRTs) were used to explore the influence of training on search performance, comparing baseline with post-training medium SFC searches, and the influence of search scale, comparing post-training medium SFC searches to post-training large SFC searches.

### 2.2. Results

Not all teams were available during all assessment periods, although 13 of the 14 teams completed the full SFC training program and 12 completed the final assessments. Despite only a brief familiarization session, group search sensitivity remained relatively high at 70% and false-alert instances low (N = 0) in medium-sized searches in the novel outdoor conditions.

SFC search sensitivity then increased from baseline to post-training assessments by approximately 10% although this was not a significant result (Figure 4). The increase in false-alert instances between the assessments (N = 0 to N = 11) was significant (z = −2.02, *p* = 0.04, r = −0.67).

Group sensitivity post-training was significantly reduced when searching a larger area compared to the medium-sized plots (Figure 4) but there were no significant differences in false-alert instances z = −1.36, *p* = 0.18, r = −0.28, with a small effect size (Cohen, 1988). Descriptive statistics on dog–handler team group performance for both search activities are presented in Table 2.

### 2.3. Summary

The aim in this experiment was to determine how well volunteer dog–handler teams, trained to search controlled indoor conditions, could search simple outdoor field conditions, immediately after a single familiarization session and then after 12 weeks of training. It was found that despite searching in a novel environmental context after only a brief familiarization session, group search performance was relatively well maintained with 70% sensitivity and no false alerts. We expected that search sensitivity (i.e., proportion of targets found relative to the total number of targets available) [33] would increase with training while false-alert instances decrease. While sensitivity did increase from 70% to 81.94% this was not a significant result. Unexpectedly an increase in false-alert instances was recorded. We also expected that, after training, teams searching an area eight times larger than they were familiar with would display reduced search sensitivity and precision. A significant decrease in sensitivity was observed in the larger search size, but no significant change in false-alert instances was recorded.

Because these results were largely contrary to expectations, they are challenging to explain. However, we anticipate that the high (70%) group sensitivity in baseline searches may have been inflated by practice effects, as all teams conducted several familiarization trials, including a 10 m × 10 m search, immediately prior to the baseline assessment. This was done to help give dogs and handlers a basic understanding of how to search in the new environmental context. While this may have somewhat inflated performance in the new conditions, it also highlights that teams were able to perform relatively well in the new environmental context after only a brief familiarization session. We further contend that it is unrealistic that many experienced CDD trainers/handlers would conduct searches of a novel environment without at least some level of familiarization and consider the board search activities used in this experiment to represent a minimal level of familiarization.

With respect to the finding that false-alert instances increased with training experience, this can be attributed to two teams (#11 and #13) who were responsible for seven of the 11 false alerts. The expected reduction in search sensitivity when search scale was increased is likely due to teams finding it more difficult to apply appropriate search strategies to search larger plots. However, in this experiment, search size was confounded with search familiarity in that the teams had been trained to search medium-sized plots but encountered a large plot only during the assessment session. This issue was partially addressed in a second experiment.

## 3. Experiment 2: Complex Field Conditions (CFC)

The primary aim in this experiment was to ascertain whether volunteer dog–handler teams, experienced in searching a SFC, would be able to work in a more complex field condition (CFC). We were also interested in documenting whether the teams showed reduced search sensitivity and increased false-alert instances when searching large areas compared with medium-sized search areas, when both were equally unfamiliar. Thirdly, we wanted to determine whether seven weeks of training in the CFC would lead to significantly increased search sensitivity and decreased search time and false-alert instances. Finally, we were interested in examining individual performance differences between our dog–handler teams.

### 3.1. Materials and Methods

#### 3.1.1. Participants

Thirteen of the 14 dog–handler teams that participated in experiment one (Table 1) were included in experiment two. One team (#10) was unable to participate due to unprovoked dog–dog aggression.

#### 3.1.2. Materials

The training materials from Experiment 1 were used, except that training and assessments took place in box-ironbark woodland in medium 25 m × 25 m (625 m^2^) and large 50 m × 100 m (1000 m^2^) areas. These areas featured a largely open understory and a small topographical gradient (Figure 5), and were used to represent a Complex Field Condition (CFC).

#### 3.1.3. Procedure

##### Familiarization

As in Experiment 1, environmental familiarization was conducted using a scent board. This process was faster and required fewer repetitions at each stage, perhaps because the teams were already familiar with searching outdoors. Most teams were able to search small areas off the board within one approximately 15 min of training (approximately five to 10 reinforcement instances in total). As with SFC sites, CFC sites contained many naturally occurring scents, including kangaroo scats, which served as non-target odors that the dogs were required to ignore.

##### Training Methodology

Teams trained over a seven-week period which included six two-hour classes and one, two-day workshop with the same structure as Experiment 1. Once each team was able to search off the board, baseline searches, as described below, were conducted and assessed. This was followed by several weeks of training, which focused on developing and implementing appropriate search strategies to cover a large bushland area (up to 5000 m^2^). This included understanding how thick vegetation and obstacles such as rocks and logs can manipulate the movement of odor through a landscape. It also included training handlers how to better recognize changes in dogs’ body language when they have detected the target odor, so that they can give dogs extra space and time to follow an odor to the source. Obedience training, including recall and emergency stop exercises, were also incorporated into training. Training and assessments were conducted with dogs both on and off long leads (approximately 10 m in length) depending on owner preference and training progress.

##### Search Assessments

Search performance was assessed twice. The first assessments followed the familiarization process and included two medium (25 m × 25 m; 625 m^2^) CFC baseline searches, with a time limit of 10 min each. After a break of approximately one hour, teams completed a large (50 m × 100 m; 5000 m^2^) CFC search with a time limit of 25 min. After seven weeks of training in medium and large CFC searches, the performance of each team in two large CFC searches was assessed again, to determine the effect of training on performance. These were conducted over two non-consecutive days, on average 7 days apart. As in Experiment 1, medium baseline assessments were distributed over eight plots, but all large searches were conducted in the same area, with different target locations being used in all searches to minimize cueing. Teams had two breaks of 30–45 min between trials and did not observe other teams’ searches. All other aspects of assessment methodology were the same as Experiment 1, including in the putting out of targets and in the use of non-target control tips and pipes in each search.

##### Data Analysis

Search performance was again measured with sensitivity (i.e., proportion of targets found relative to the total number of targets available) [33], false-alert instances and search effort (i.e., time spent searching an area per search) [33]. We also report search precision (i.e., the proportion of alerts directed towards a true target) [33]. Paired-samples t-tests were used to compare search time data when these were normally distributed. Kolmogorov-Smirnov tests revealed that search sensitivity and false-alert instances were not normally distributed due to predominately correct responses. Therefore, as advised by Pallant [34], non-parametric statistics were used for analyses involving these variables. Wilcoxon Signed Rank Tests (WSRTs) were used to explore the influence of environmental context on search performance, comparing medium-sized SFC searches to medium-sized CFC searches. WSRTs were also used to explore the influence of search scale on search performance, comparing performance on medium and large CFC searches, and to explore the influence of training on performance, comparing large CFC searches before and after 7 weeks of training. Individual differences in teams’ performances were examined at a descriptive level only.

### 3.2. Results

Descriptive statistics for group performance are presented in Table 2. WSRTs between medium SFC post-training searches (conducted at completion of Experiment 1) and medium CFC baseline searches (conducted at beginning of Experiment 2) revealed significantly poorer search sensitivity in the CFC (Figure 6) despite teams still locating 60% of targets. WSRTs also revealed a significant difference in the mean number of false alerts (N = 11 to N= 0), z = −2.03, *p* = 0.04, r = −0.43, (medium effect). On average, teams also spent more time searching in the medium CFC than medium SFC. A paired-samples t-test indicated this difference was not significant (t (9) = −1.98, *p* = 0.08), although a large effect size of η^2^ = 0.30 (Cohen, 1988) was found. In addition, the baseline CFC searches revealed considerable variability in search sensitivity scores between dog–handler teams, ranging from 33% to 100% of targets (data not shown).

WSRTs used to explore the influence of search scale on performance in a novel, complex environment found no significant difference in search sensitivity between medium and large CFC searches (Figure 6), but did detect a significant increase in mean number of false alerts z = −2.43, *p* = 0.02, r = −0.73 in the larger area.

WSRTs revealed a significant improvement in search sensitivity for large CFC sites between baseline and post-training assessments (Figure 6). No significant differences were found between baseline (N = 3) and post-training (N = 6) false-alert instances, z = −1.29, *p* = 0.20, r = −0.25 (small effect) or search effort (baseline mean = 21 min, 14 s; post-training mean = 23 min, 17 s), and z = −0.71, *p* = 0.48, r = −0.14 (small effect) (Cohen, 1988). Similarly to medium CFC searches, a high level of variability was observed, with sensitivity ranging from 29% to 100% between teams and differences of up to 43% within teams between Searches 1 and 2. These data are presented in Table 3. Across all searches, six (46%) of the 13 teams found at least 75% of targets within 25 min.

### 3.3. Summary

The aim in Experiment 2 was to assess performance of 13 volunteer dog–handler teams in complex field conditions (CFC), before and after a seven-week training program. It was expected that teams would initially display reduced search sensitivity and increased false-alert instances and search effort (time spent searching per search) in a medium size search area compared to their post-training performance in simple field conditions (Experiment 1). While teams still found 60% of targets overall in the new context, the data were largely consistent with expectations. The increase in search effort was not statistically significant even though the effect size was large, suggesting insufficient power. It was also expected that, in this novel environment, teams would show reduced search sensitivity when searching a large size area compared to a medium-sized area. There was no significant difference in search sensitivity but increased false-alert instances were observed which may suggest that some aspects of search performance can generalize to new contexts more readily than others. Finally, it was expected that seven weeks of training in complex field conditions would result in significantly increased group search sensitivity and decreased false-alert instances and search effort in large complex field condition searches, when compared to baseline search performance. This was partially supported in that post-training assessments indicated a significant increase in sensitivity while changes in false-alert instances and search effort were non-significant. It is noteworthy that overall weather conditions during large CFC post-training assessments showed some differences to baseline searches and it is possible that these influenced search performance to some extent. It was also noted that there were large individual differences in teams’ performance.

## 4. Discussion

In this study, we aimed to explore how well volunteer dog–handler team search performance acquired in a safe, well-controlled indoor environment generalizes to searches in unfamiliar outdoor environments. We also investigated whether the scale of the search size is pertinent, and whether additional training in the field, designed to help teams develop skills relating to dog handling and designing and maintaining appropriate search strategies, improves performance. The results were complex but, in general, the search performance of dog–handler teams demonstrated moderately good context generalization. Teams previously trained entirely indoors where they found 94.12% of targets were able to locate 70% of targets in a medium size simple field condition (SFC), with only a small period of familiarization to this new context. Following training in the SFC, they were able to find 60.61% of targets on their first exposure to a more complex field condition (CFC), again following just a very short familiarization process in these conditions. Nonetheless, as expected, performance decreased whenever the scale or environmental context of a search differed from what they were familiar with. Our results were not always statistically significant, most likely due to small numbers and the challenges of working in the field with inexperienced handlers and dogs. However, when either the search scale or environmental context were unfamiliar, we observed a reduction in search sensitivity of approximately 20% compared to the next most familiar context. After training in each new context, group search sensitivity then increased by 10–20% compared to baseline performance. This highlights that the stepped training approach we have implemented is effective. Training dogs in a laboratory can prepare them well for working outdoors by developing foundational skills, although accounting for the search scale and environmental context that teams will be working in should be considered an important second step when training and assessing CDDs. Context generalization is clearly an important component of CDD training.

Dog–handler teams that were trained to search simple outdoor field conditions displayed significantly reduced search performance in areas that were larger than they were familiar with, even in the same environmental context. While false-alert instances in larger searches were not significantly different from the medium searches, teams did find significantly fewer targets in the larger search context, despite the environmental conditions remaining constant. In order to account for any practice effects arising from assessing the influence of search scale on performance in SFC, we examined the influence of search scale in an unfamiliar CFC, which again revealed a decrease in search sensitivity and increase in false alert rate. Even though teams’ ability to locate targets in novel contexts was not lost entirely and teams still found at least 50% overall sensitivity in baseline searches throughout the study, these results support our first hypothesis that searching in an unfamiliar scale would negatively impact search performance. This reduction could be explained by teams finding it more difficult to apply and maintain appropriate search strategies to search larger plots as comprehensively as the medium plots they were familiar with, within the time limit. Future research and development of CDD teams would benefit from training and assessing teams in similar-sized areas to those they will be working in.

Assessing search performance in a bushland environment that was more complex than dog–handler teams were familiar with led to decreased search performance. Teams recorded significantly lower sensitivity and more false alerts when searching in comparatively complex box-ironbark bushland after only training on a simple, open sports field. This is likely attributable to the variable topography and increased vegetation characterizing the more complex conditions, which influence how scent moves through an environment [18,23,24], the ability to maintain effective search strategies and in turn, the difficulty of searching in CFC compared to SFC. While teams still managed to find 50% of targets in large baseline searches in CFC with minimal familiarization training, search sensitivity was significantly improved after gaining seven weeks’ search experience compared to baseline assessments, supporting our second hypothesis that training would improve search performance in a given context. While teams were allocated 25 min to complete large CFC searches compared to 20 min in SFC, it is likely that further increasing maximum search effort (i.e., time spent searching) would improve sensitivity [16,33]. As two teams found 100% of targets during baseline large SFC and post-training large bushland searches within the allocated time limits, and many handlers chose to conclude the search within the time limit, it seems it was possible to search the area comprehensively within these constraints. Furthermore, implementing search time limits allows influences on search performance to be explored, such as individual dog–handler team learning rates and performance characteristics (Table 3). The average weather conditions during large CFC post-training assessments were 11–13 °C warmer and 28% less humid compared to large CFC baseline searches. It is possible that the increased temperature made odor targets more volatile [17,18] and therefore easier to detect, however, previous research has indicated that temperature may not always influence the number of targets found during field searches [16,18,35,36,37]. It was observed, however, that dogs’ respiration rate (i.e., panting) increased more rapidly during the warmer post-training conditions, particularly when searching in direct sunlight than in the cooler baseline searches. Handlers and experimenters occasionally requested longer breaks between searches to ensure dogs maintained comfortable body temperatures which indicates the warmer temperatures had some influence on teams’ searching. Considering the influences of increased complexity and unfamiliar environmental conditions on search performance, future research and development of CDD teams would benefit from training and assessing teams in environmental contexts similar to those that they will be working in.

Aside from dogs’ odor detection capabilities, the implementation of appropriate CDD search strategies for the target type (e.g., scat, nest), search context (e.g., environment and scale), search aims (e.g., determine occupancy vs abundance), sensitivity requirements and weather conditions, are likely to influence search effort and effectiveness [28]. Target odors (and un-scented controls) were placed in pseudo-random locations and equally distributed across all areas of the search plot. This is in contrast to some CDD search applications in which search effort can be focused on areas with a greater probability of detection, such as those with certain habitat characteristics preferred by a target species [28]. As search scale increases, teams are required to travel further, expending more effort and energy than in smaller areas, which can increase the importance of effective and efficient search strategies, particularly when teams conduct multiple searches in a day. Future research into CDD search strategies and search efficiency in various weather and environmental conditions CDDs is recommended [28].

Training in this study was tailored to the needs of each team, although all teams were trained using the same overall methods. A greater range of performance between CDD teams was observed in CFC searches than in SFC searches. In the two post-training CFC searches conducted by each dog–handler team, no significant group performance differences were present between searches, but large individual differences were present, with three teams displaying a 43% (3 targets) difference in search sensitivity. This did not appear attributable to weather conditions; there was little difference in temperature and humidity between these assessment periods, although slightly stronger (up to 7 km/h difference) wind gusts were recorded during Search two, which may have influenced target detectability somewhat. The overall sensitivity between teams ranged considerably, from 28% to 86%. However, despite this variability, approximately half of all teams found almost 80% of the targets after seven weeks of training. We anticipate more teams would reach at least 80% after 12 weeks of training, but acknowledge that our diverse group of volunteers includes teams with different performance characteristics, that may be more or less suited for particular detection roles. In addition to search performance, experimenters anecdotally observed decreases in the confidence of some handlers, particularly around their ability to maintain appropriately spaced and parallel transects when searching in bushland. This was particularly so during baseline searches, but was also observed in some teams during post-training assessments. While post-training assessment data were collected over at least two searches, assessing dog–handler team search performance over multiple searches is recommended to gain a reliable representation of performance. Here, the potential influence of different weather conditions on performance over different assessment periods should also be considered.

Training multiple CDDs is time and resource consuming, so much of the training and search performance literature on CDDs involves a relatively small sample. In fact, a recent review by Bennett et al. [33] of 61 studies that reported quantitative information on the performance or cost of detection dogs involved in conservation projects revealed that the average number of dogs used in the 57 studies reporting dog sample size was four. While the importance of small sample studies and the relevance of their findings should not be undervalued, a strength of our study is that the sample of 13 dogs is considerably larger than most others. Furthermore, the diversity of dog breeds and handler characteristics in the sample is likely to be a good representation of community owners and pet dogs that might participate in future volunteer-based CDD programs. This is in contrast to findings from studies with restricted variation in breed or handler characteristics. Despite the extensive amount of data collected, there are limitations in this study that are worth noting. Firstly, targets in this study were placed by experimenters up to 20 min before training and search assessments, meaning the improvements in sensitivity in post-training searches compared to baseline performance may be partially explained by dogs learning to track experimenters to target locations. While no instances of dogs tracking humans to target locations were noticed in this study, efforts were made to reduce this by experimenters walking indirectly in zig-zag or loop patterns to and from target locations when placing targets out for search assessments. Furthermore, as search areas contained numerous human tracks from experimenters and handlers, any instances of dogs following human tracks to locate targets are likely to be a minimal influence on overall results. Further attempts to account for human scent indicating target presence may be useful in future research (e.g., placing out targets further in advance or with drones), particularly when working with subtle target odors. A second limitation relates to target density. Previous research has indicated that expectations of target density held by both dogs and handlers can influence search performance [17,38] and teams in this study were therefore blind to the number of targets available in each search assessment. Furthermore, the number of targets per search was often changed throughout this study in an attempt to prevent the development of expectations of target numbers, which could influence search behavior. However, we were not able to control for target density per area in this study, meaning the probability of finding a target by chance ranges from one target per 100 m^2^ (Small SFC baseline) to one target per 1250 m^2^ (Large SFC baseline). Within the trials compared statistically, the difference in target density ranged from no difference (comparison between Large CFC baseline and Large CFC post-training assessments) to one target per 937 m^2^ (comparison between Medium SFC post-training and large SFC baseline). This presents a limitation to interpreting some findings in this study. We tried to balance ecological validity against rigorous experimental design but we acknowledge that the relevance of target density differences between searches should be considered in future research on search context and performance. However, it is noteworthy that in the application of CDDs to detect conservation related species, the density of many targets shows marked variation across areas when variables such as species habitat and food preferences, behavioral patterns, environmental types, population characteristics and numerous other factors are unable to be controlled. This, in turn, may influence CDD search strategy and the probability of detection [28,39].

Overall, this study demonstrates that volunteers and pet dogs of numerous breeds can be trained in CDD skills to conduct area searches of varied size and environmental complexity to find scented targets that are not visually salient to handlers. Because of our focus on safely training the dogs and handlers, we used an essential oil odor, rather than a biological, conservation related target. Hence, further research on the ability of volunteer teams to work around wildlife in a safe and non-interactive way in uncontrolled field conditions is needed. Also relevant to operational CDD search performance is dogs’ ability to perform olfactory generalization, such as generalizing from training samples to field samples of conservation related odors, which may smell different due to diet or other environmental factors [22]. While further research is required to understand volunteer CDDs abilities in these areas, four teams in this study subsequently participated in field survey deployments searching for the endangered Alpine stonefly (*Thaumatoperla alpina*). Here, all teams present located numerous individuals [20] and the teams displayed generalization to a similar yet different species. This indicates that volunteer teams are capable of detecting conservation related targets in the field and can display evidence of generalization between training samples and naturally occurring samples, in addition to species level generalization. Future research could also explore differences in the rate that false alerts are made towards non-target controls and other naturally occurring objects/odors encountered during searches.

While a volunteer-based model of CDD training and deployment may help increase access to skilled teams by reducing financial costs, these are not negated entirely. This study was achieved at the financial cost of engaging two professional trainers to conduct group training sessions. As this was an experimental study involving a considerable amount of data collection over multiple training and assessment phases which involved professional trainers, the timeframe and costs of our program may be longer and more substantial than one solely aiming to develop operational teams. Overall, this study resulted in a pool of volunteer dog–handler teams experienced in CDD training and detection skills in different environments that can commence training on biological, conservation related targets. We anticipate search performance will continue to improve with further search training on such targets in the relevant scale and environmental contexts. This may also lead to improvements in the search speed and search stamina of volunteer teams. However, volunteers would likely not be required to search as quickly or for as long as professional teams, particularly as operating as a group allows teams to rotate and rest in between searches. Furthermore, we recommend engaging experienced professional trainers to supervise and coordinate volunteer teams conducting field deployments to help ensure teams can operate in a safe and effective way. While this would incur additional costs, one trainer can supervise more than one volunteer team at once, allowing multiple teams to be searching and resting at the cost of engaging one trainer. In this way, a volunteer-based model can assist in making multiple skilled CDD teams more accessible to conservation projects [11,12]. However, as some level of volunteer attrition during training or deployment periods is almost unavoidable, [40] the longer-term cost savings of training and deploying volunteers must be balanced with the degree of participant dropout and the costs of training new volunteers to maintain a viable pool of teams. In this way, understanding volunteer motivation and satisfaction is important to recruiting committed teams and promoting long term engagement and warrants further investigation. While it is unlikely that a volunteer-based model of CDD training and deployment will replace the need for professional CDD teams, the model may help achieve important conservation outcomes in suitable applications at a greatly reduced cost.

## 5. Conclusions

This study demonstrates that dog–handler teams in a volunteer-based conservation detection dog (CDD) model can be trained to successfully search different field conditions. Search performance in bushland conditions varied within our cohort of dog–handler teams but included high performing teams with at least 80% search sensitivity. While the performance of teams trained in a context with familiar search size or environmental conditions showed moderately good generalization to unfamiliar contexts with sensitivity ranging from 50 to 70%, teams displayed approximately a 20% decrease in sensitivity in all baseline assessments conducted in unfamiliar contexts. However, sensitivity improved by around 10–20% after additional training in the new context. Future research, development and assessment of CDDs should include training teams in environmental conditions and at a search scale that is similar to those they will be working in. Further research should also explore the utility of volunteer teams to detect conservation related odors in uncontrolled field conditions and explore the influences of search strategy on performance in various conditions.

## Figures and Tables

**Figure 1 animals-11-01177-f001:**
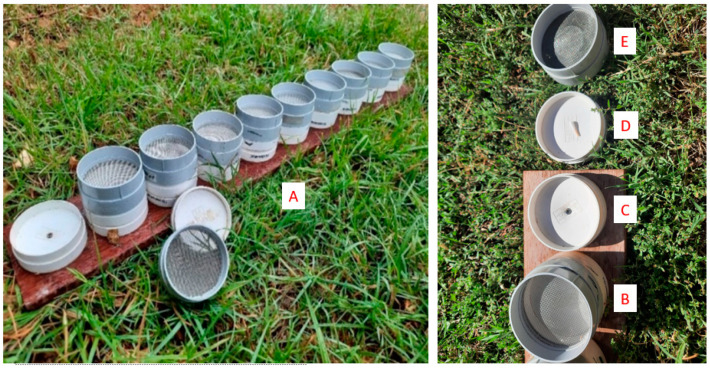
Scent board with 10 detachable PVC pots used in training (**A**). Each complete pot (**B**) consists of a base fitting (**C**) screwed to a timber board, a detachable tray fitting (**D**) into which an odor can be placed and a mesh cap fitting (**E**) which prevents access to the odor.

**Figure 2 animals-11-01177-f002:**
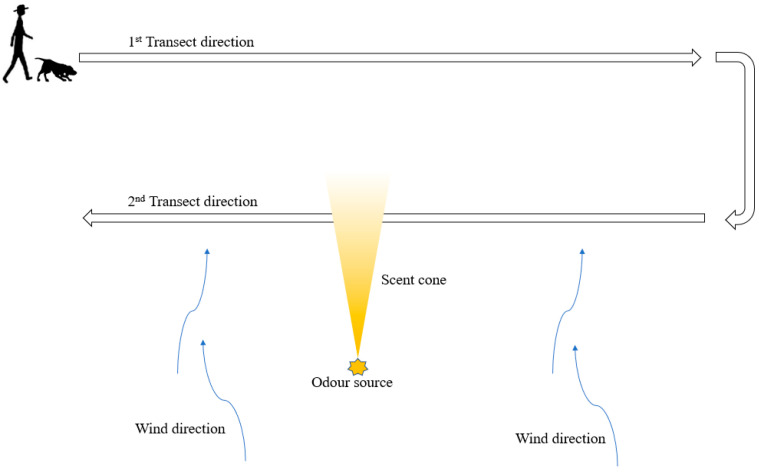
Second stage of transect search training designed to help increase search scale and search stamina. Here teams begin downwind from the target odor and commence an initial transect outside of a scent cone before progressing through it on subsequent transects.

**Figure 3 animals-11-01177-f003:**
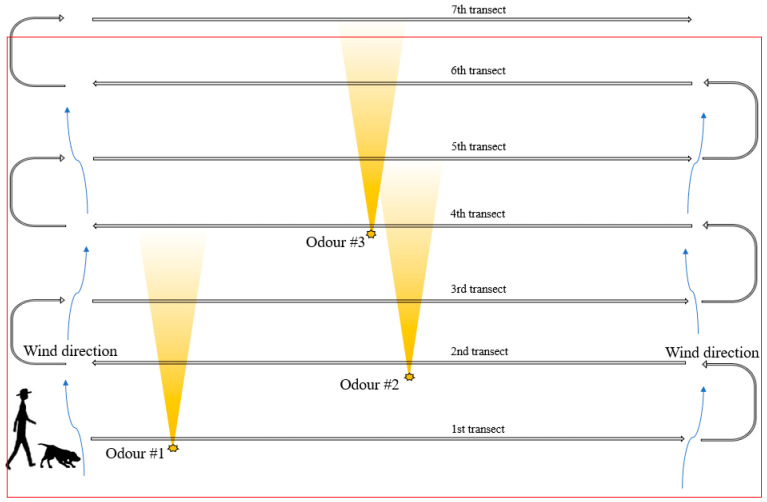
Third stage of transect search training designed to help increase search effectiveness and area coverage. Here teams begin upwind from the target odor and complete parallel search transects, enabling dogs to sample a new ‘corridor’ (area between transect lines) of odors with each new transect.

**Figure 4 animals-11-01177-f004:**
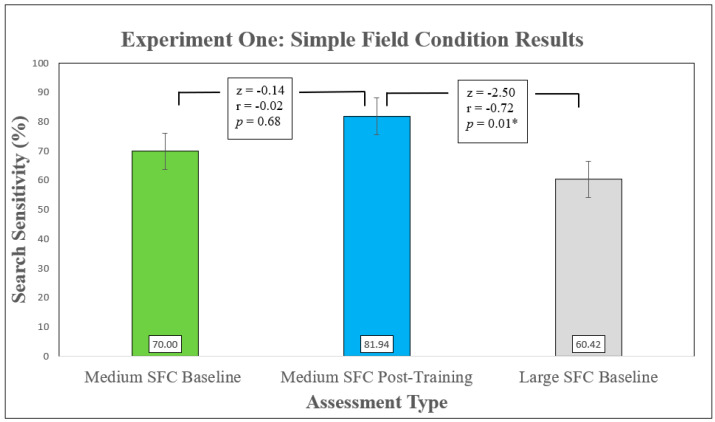
Group search sensitivity over the three different Simple Field Condition (SFC) search assessments of Experiment 1 are presented in chronological order: Medium SFC baseline (N = 10), medium SFC searches after 12 weeks of training (N = 12) and large SFC baseline searches (N = 12). Results of Wilcoxon Signed Rank Tests between each assessment are also presented. ‘*’ Indicates statistical significance (*p* ≤ 0.05). Error bars represent standard error.

**Figure 5 animals-11-01177-f005:**
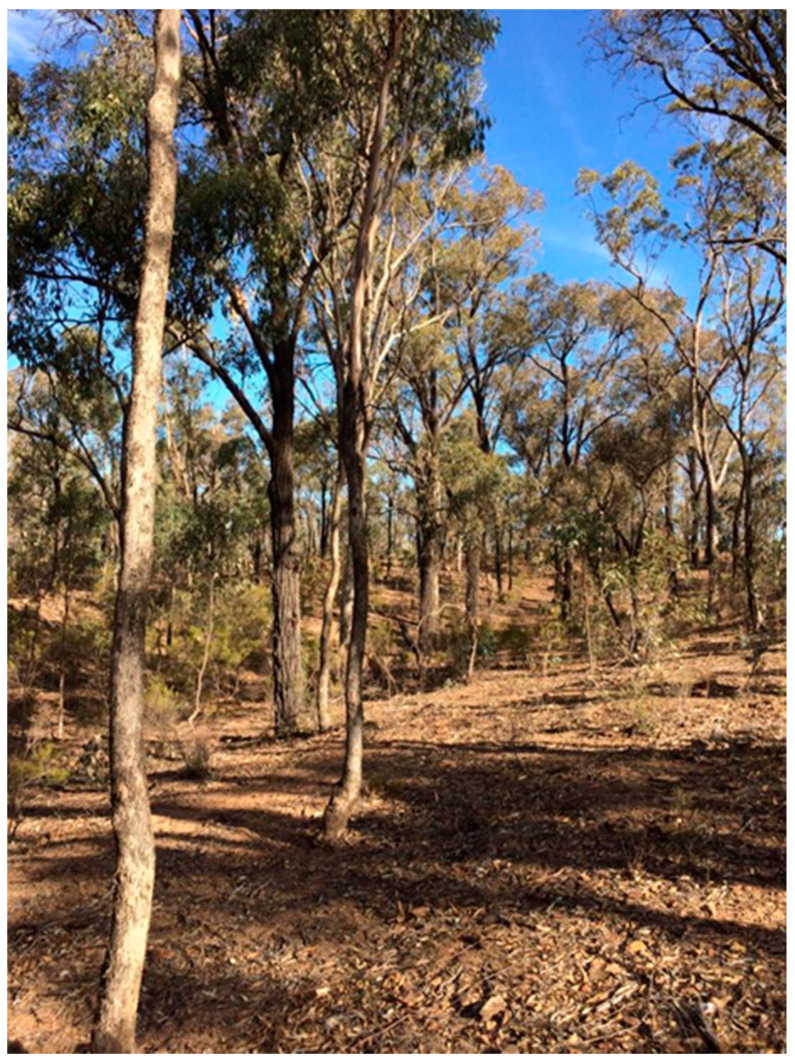
Box-ironbark woodland used to represent a complex field condition (CFC) during training and assessments.

**Figure 6 animals-11-01177-f006:**
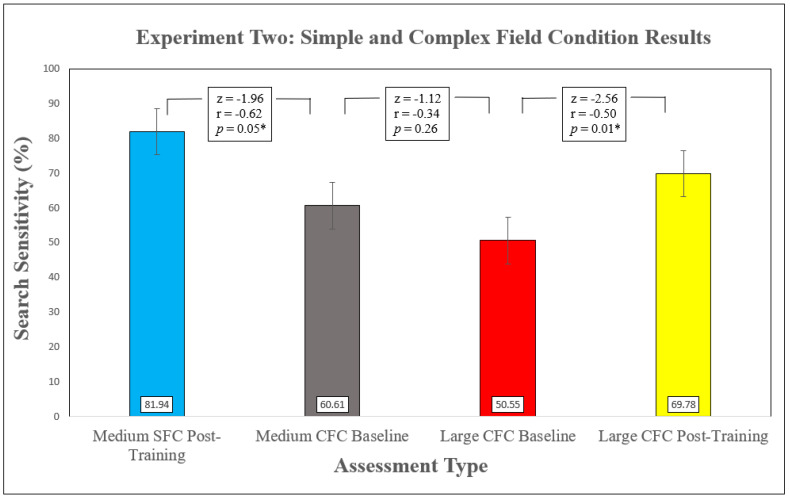
Group search sensitivity over the four different search assessments of Experiment 2 are presented in chronological order: Medium Simple Field Conditions (SFC) post-training (N = 12), medium Complex Field Conditions (CFC) baseline (N = 12), large CFC baseline (N = 13) and large CFC post-training (N = 13). Results of Wilcoxon Signed Rank Tests between each assessment are also presented. ‘*’ Indicates statistical significance (*p* ≤ 0.05). Error bars represent standard error.

**Table 1 animals-11-01177-t001:** Dog–handler team demographic information at the start of the current experiment.

Dog–Handler Team Number#	Owner-Reported Dog Breed	Dog Sex	Dog Age (years)	Handler Gender	Handler Age (years)	% of the 19 Training Sessions Attended (No of Sessions)
#1	Rhodesian Ridgeback	Intact male	5.8	M	39	74% (14)
#2	Australian Kelpie	Intact male	5.7	F	58	79% (15)
#3	Miniature Poodle	Neutered female	2.5	F	60	90% (17)
#4	Samoyed	Neutered female	2.2	F	34	74% (14)
#5	Rough Collie	Neutered male	6.6	F	63	95% (18)
#6	Cocker Spaniel × Toy Poodle	Neutered male	2.1	F	29	58% (11)
#7	Weimaraner	Intact male	4.9	M	54	74% (14)
#8	Labrador × Kelpie	Neutered male	3.3	F	31	63% (12)
#9	Cavoodle	Neutered male	1.8	F	50	68% (13)
#10	Australian Cattle Dog	Neutered female	7.3	F	65	50% (6/12) *
#11	Border Collie	Neutered female	4.7	M	68	58% (11)
#12	Border Collie	Neutered female	3.8	F	25	74% (14)
#13	Border Collie	Neutered female	8.6	F	67	68% (13)
#14	Labrador	Neutered male	2.7	F	37	37% (7)

* Dog–handler team did not participate after 12 weeks due to unprovoked dog–dog aggression.

**Table 2 animals-11-01177-t002:** Combined results for standard room searches, taken from Rutter et al. [12], Simple Field Condition (SFC) baseline searches, SFC post-training search assessments, Complex Field Condition (CFC) baseline searches (Experiment 2) and CFC post-training searches (Experiment 2). Sensitivity refers to proportion of targets found relative to the total number of targets, precision refers to the proportion of alerts directed towards a true target and search effort refers to search duration per search [33].

Assessment Number and Search Size	N Dog–Handler Teams	N Trials per Team	N Targets per Search	Targets Available for Analysis *	Total Correct Alerts	Grand Mean Sensitivity (%) **	Total False Alerts	Search Precision	Search Effort (Time) (Min:Sec)			
									Mean	SD	Min	Max
PREVIOUS ASSESSMENT: Controlled indoor conditions Post-Training. Weather data not collected.												
Standard room search	17	1	1	17	16	94.12	1	0.94	0:21	0:18	0:06	1:25
ASSESSMENT 1: Simple Field Conditions Baseline. Weather data not collected.												
Small	10	1	1	10	6	60.00	2	0.75	1:18	1:41	1:09	2:20
Medium	10	1	1	10	7	70.00	0	1.00	2:38	2:02	0:20	5:00
ASSESSMENT 2: Simple Field Conditions Post-2 weeks of training in Medium plots. Temperature: 6 °C–12 °C (M = 8 °C). Humidity: 55–97% (Mean = 65%).												
Medium	12	3	2	64	54	81.94	11	0.83	5:17	1:58	1:30	9:53
Medium (Target absent	12	1	0	0	-	-	4	-	5:28	2:20	1:35	10:00
Large	12	1	4	48	29	60.42	2	0.94	16:30	2:28	13:25	20:00
ASSESSMENT 3: Complex Field Conditions Baseline. Temperature: 3 °C–13 °C (M = 10 °C). Humidity: 47–97% (Mean = 64%).												
Medium	12	1	3	36	22	60.61	0	1.00	7:32	2:26	4:00	10:00
Medium (Target absent)	11	1	0	0			1	-	6:34	1:46	3:08	8:52
Large	13	1	7	91	46	50.55	3+	0.94	21:14	3:52	15:02	25:00
ASSESSMENT 4: Complex Field Conditions Post-seven weeks of Training in Medium and Large plots. Search 1: Temperature: 16 °C–25 °C (M = 21 °C). Humidity: 28–45% (Mean = 36%). Wind conditions: Occasional gust 0–6 km/h. Search 2: Temperature: 20 °C–25 °C (M= 23 °C). Humidity: 29–48% (Mean= 36%). Wind conditions: Occasional gust 0–13 km/h.												
Search 1: Large	13	1	7	91	61	67.03	5+	0.92	22:55	2:41	17:13	25:00
Search 2: Large	13	1	7	91	66	72.53	7++	0.90	23:38	1:52	19:73	25:00
Combined total	13	2	7	182	127	69.78	12	0.91	23:17	2:18	17:16	25:00

* This column refers to the N of targets available for analysis after accounting lost data. ** This column refers to the grand mean of the search sensitivity of all teams, rather than average sensitivity. Note: Controlled indoor conditions standard room search from Rutter et al [12] encompass 16.8 m^2^, Small searches encompass 100 m^2^, Medium searches encompass 625 m^2^ and Large searches encompass 5000 m^2^ for all outdoor conditions.

**Table 3 animals-11-01177-t003:** Large (5000 m^2^) complex field condition search assessment results for each dog–handler team after seven weeks of training. Shaded rows indicate teams that found a total of at least 75% of targets across both assessments. Sensitivity refers to proportion of targets found relative to the total number of targets available, effort refers to search duration per search, precision refers to the proportion of alerts directed towards a true target [33].

	Search 1			Search 2			Searches 1 and 2 Combined		
Dog–Handler Team No.	Search Sensitivity % (Total = 7)	Total False Alerts	Search Effort (mm:ss)	Search Sensitivity % (Total = 7)	Total False Alerts	Search Effort (mm:ss)	Total Search Sensitivity %	Total Search Precision	Mean Search Effort (mm:ss)
#1	57.14 (4)	1	25:00	85.71 (6)	1	24:26	71.43	0.83	24:43
#2	57.14 (4)	0	21:32	85.71 (6)	0	19:43	71.43	1.00	20:37
#3	28.57 (2)	0	23:06	28.57 (2)	1	25:00	28.57	0.80	24:03
#4	57.14 (4)	0	24:22	100.00 (7)	0	25:00	78.57	1.00	24:36
#5	71.43 (5)	1	25:00	85.71 (6)	1	24:44	78.57	0.85	24:52
#6	42.86 (3)	1	20:29	85.71 (6)	0	21:54	64.29	0.90	21:11
#7	42.86 (3)	0	23:51	71.43 (5)	1	24:19	57.14	0.89	24:05
#8	85.71 (6)	1	25:00	71.43 (5)	1	25:00	78.57	0.85	25:00
#9	85.71 (6)	1	25:00	42.86 (3)	-	-	64.29	0.86	-
#11	100.00 (7)	0	25:00	57.14 (4)	-	-	78.57	1.00	-
#12	85.71 (6)	0	17:13	85.71 (6)	0	21:36	85.71	1.00	19:24
#13	85.71 (6)	0	18:24	57.14 (4)	0	23:13	71.43	1.00	20:38
#14	71.43 (5)	0	24:04	85.71 (6)	2	25:00	78.57	0.85	24:32
Mean	67.03 (61)	0.34	22.55	72.53 (66)	0.54	23.38	69.78	0.91	23:17

## Data Availability

Not applicable.

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
