# Peer review of "Diving in Nose First: The Influence of Unfamiliar Search Scale and Environmental Context on the Search Performance of Volunteer Conservation Detection Dog–Handler Teams"

_animals, 2021, doi:10.3390/ani11041177_

Round 1
Reviewer 1 Report
This article is a continuation of work that has been carried out in a controlled context (indoor room) with volunteer human-dog teams (pet dogs). The overall project aims to determine whether it is possible to recruit, train and work with these teams from an environmental (endangered plant and animal species) protection perspective.
This manuscript presents two experiments that were conducted outdoors, on a 'simple' field (sports field) and on a 'complex' field (woodland).
The article is well written, its protocol well constructed and the results intelligently discussed and open up many perspectives.
Suggestions for improving this manuscript are as follows and relate mainly to the description of the protocol and the sequencing of the stages and test sessions:
Text :
- L109/303: to complete titles. Experiment 1 : Simple Field Conditions (SFC), Experiment 2: Complex Field Condition (CFC)
- Materials section (L118): What protection do odour handlers wear? (gloves?)
- Procedure section (L133): please describe classes : training time duration per teams/dog, and workshops: timing/content.
- L147 : FamiliArisation
- L 153 and 157 : please provide a number / a range
- L157 : full scent board with no odor cue ?
- Training methodology (L171) : original transect, new transects, scent cone … : a figure will help to understand.
- L183 and 186: Are false alerts only suspected to be due to residual odor ? Isn’t it disturbing to the dogs ? Please discuss this.
- L 193: small and medium sized SFC not clearly described in the Procedure section.
- L203: only 2 searches per plot were conducted but according to which unit of time? 2 searches per team or 2 searches per day or…
- L211-212: Frequency of target use ? Single use ?
- L217-18 : This information is very important but is lost in the text. It should appear earlier, as well as special attention should be given to the test ‘odor absent’ L229-231 (négative control test ?). What about its frequency and its location in the whole procedure (see below, description of sessions) ?
- -L236: search effort is missing (but is mentioned in Table 2): please add and explain (everywhere in the text such as in Table 4 caption) what ‘search duration’ means (Is it ‘search duration per search’)?
- L317 and Table 1 footnote : specify dog-dog or human-dog aggression or if that dog had been aggressed by another dog.
- L321: woodland: size ?
- L336: Did dogs also worked on/off leads ?
- L539: previous section section talked about operational CDDs (L527) and this section talks about ‘volunteer’ CDD teams. Please start the sentence: In this research/In this work
Tables - Figures
- Figure 1: one pot seems to have three levels. An arrow could usefully locate them and the legend could explain them (grid, smell =black spot on the photo? , intermediate level).
- 2-3-5-6-7: these figures are not essential and the results of the statistical tests could appear in tables 2 and 3 or in a new table (signif. comparisons results or effects)
- Although two experiments were conducted, the second one is a continuation of the first and takes into consideration some of its results. Therefore, it is advisable to merge Tables 2 and 3 (note: Table 2-small room searches data: please indicate their reference because not collected in this paper or mention them in Search assessments L187. Remind dimensions in footnotes). In the same vein, a timeline could usefully illustrate the course of a human-dog team (intervals between measurements, breaks between sessions) and indicate when the test sessions are carried out (=data collection). For example, there is a pre-training assessment (L170) but what are the other assessments, when do they take place (training level ?), what are the field conditions, duration, number of targets, intervals ... (L188: not true that there are 3 assessments, see Table 2, first column where ‘indoor’ results are taken into account. L200: Final session/assessment: interval from the previous one ?)? A number of 20 training sessions was estimated but is this correct? This precision could be added in the protocol or in Table 1 (last column). When does the odor-absent session occur ?
- Table 4 caption : please add : assessment results ‘from each dog-handler teams under study’ after seven weeks of training
Author Response
Please see the attachment which contains reviewer one's comments highlighted in green and changes underneath.

Reviewer 2 Report
In general I think the topic is interesting and a valuable one to look at more closely. However, some of the choices that have been made in this study have made the results rather limited in scope. The familiarization for each of the two new search areas for example ruins the basic idea of generalization – how is performance in a new environment? Training first and then testing is very comforting for dogs and handlers, but really influences your data. The variation in probability of coming across a target varies by a factor 6 over the scenarios but is not evaluated as a variable. The lack of controls is also a problem: although there are odour controls in the SFC they are lacking in the CFC. Alternative explanations as to results (learning to follow human cues during training in the CFC and the higher temperature in assessment 4 compared to 3, resulting in quite a different odour availability) are missing. Besides this, I have seen one element in the data that is either wrong, or missing an explanation. Since this is a quite important number (sensitivity in assessment 2, medium search area) it puts much of what is discussed in a different light. If it is wrong it might require some re-work in the statistics, if it is missing an explanation it is less work. Finally, it would be good to also address odour generalization as a second part of generalization in the introduction and discussion. I am including more detailed information below.
Line 80: generalizing in detection training has two meanings – the one cited here, nl. context generalization, but the second is odour generalization. It should be made clear here that since the odour is remaining EXACTLY the same all the time, these experiments are dealing ONLY with context generalization. This makes the paper valuable but also very limited in scope. Especially since this is strictly speaking not testing context generalization since they are allowing familiarization prior to any testing.
Line 87 – since the reference list is missing, I wonder if the authors are familiar with the excellent paper Porrit et al (2015) have written on the subject? (Porritt, F., Shapiro, M., Waggoner, P., Mitchell, E., Thomson, T., Nicklin, S., & Kacelnik, A. (2015). Performance decline by search dogs in repetitive tasks, and mitigation strategies. Applied Animal Behaviour Science, 166, 112-122.)
Line 147 – I understand the need for familiarization. However it would be good to quantify this to some extent. One way of doing that is describing the number of reinforcements each dog received prior to their doing the first (pre-training) assessment.
Line 196 – another problem I see with all the variations is that the probability of encountering a target is also very different between the different scenario’s – varying from finding 1 per 100m2 to finding 1 per 625m2 with a number of probabilities in between, and no attention is being paid to this variable either in analysis or discussion.
Line 238. False alerts as calculated here are not a ‘rate’, since they are not a proportion of anything. It would provide more insight to calculate divide the false alarms in two groups: one as a ‘rate’ – number of false alerts to the control unscented tips in the search environment (or to those they actually smelled); and ‘random’ false alerts to non-introduced items/scents. You can then also compare the FA rates (so using the controls) to compare what the dogs did in the odor absent fields.
Table 2 contains an error: assessment 2, medium, has 12 teams that each conduct 3 trials where they can find 2 target. That leads to a total of 72 targets, not 64, which decreases the sensitivity to 75%. This may also have an effect on the significance calculated in Fig 3.
Line 328 – again, some numbers to show that the familiarization went faster would be interesting. How many reinforcements were necessary here?
Line 347 – the description of search assessments in experiment 1 included information on the number and type of controls put out. Do I understand here that NO effort was made to put out controls, assuming that there were ‘natural’ controls here? This is a fundamental problem. By not putting out controls, the dogs may learn during the training to pay attention to the human tracks that lead to the scented tips, since they were put out just before the search. This could be one reason for the improved detection after training.
Line 362 – again, you cannot calculate ‘false alert rates’ since they are not a proportion of anything. And since no controls were put out, you cannot divide the FA’s into anything that makes sense.
Table 3 – same error as table 2 with the assessment 2 medium search.
Also not that between assessment 3 and assessment 4 there is a HUGE difference in average temperature, which would make the samples much more easy to smell. This may be a second explanation for the improved detection (besides the additional human scent cue I discussed above).
I would combine tables 2 and 3 into one, now there is redundancy. I would also combine all the figures into one – basically there are 5 bars, and you can just illustrate which ones are significantly different in that one figure. That would provide more insight.
Line 465: As explanation for the improvement from assessment 3 large to assessment 4 large: you need to include the possibility of having learned to track human scent (that you have no controls for) and the higher temperature as explanations.
Line 482: check these numbers. On the SFC there is 5% improvement, in the CFC 20 but here we have the problem of the possible learning to track human cues as well as the significant difference in temperature as possible explanations.
Line 499: the comparison of the SFC and CFC different search scales cannot be made directly, sinc the large SFC only contained 4 targets and the large CFC contained 7. This may have influenced the results significantly, since the dogs are rewarded after every find. If you re-calculate based on the mean time spent searching, and the number of targets that were found, on average it took 6:45 in the SFC and about 6min in the CFC. This may just have been their speed of working.
I would include some comments on odour generalization to also be considered in training CDD. Depending on the species being searched for, differences in diet and environment may mean changes in the odour that the CDD have to be looking for. There is a nice review paper on this by Moser (Moser, A. Y., Bizo, L., & Brown, W. Y. (2019). Olfactory generalization in detector dogs. Animals, 9(9), 702.)
Author Response

(The authors gave the same response as above.)

Round 2
Reviewer 2 Report
Much has been clarified, also the other reviewer raised some very relevant points that have greatly helped to clarify the content of the paper. Three comments:
In professional police, customs, military detection dog training, context generalization is part of training. In such organisations, it would never be accepted that dogs are trained ONLY in a training center and then immediately deployed. Either it is part of training, or there is a period where the dogs go through a ‘being trained on the job’ before they are deemed mission ready. It may be wise to point this out somewhere – and I fully accept that this is not as common in volunteer detection dog programs.
Lines 277-280: controls that were put out allowed experimenters to monitor the false alerts. However, it seems they did not and nowhere is there a reference to why not, only in accompanying letter where the authors state that this information is not available? This is a serious error in data collection and needs to be commented on in the paper.
Line 658-668: Variation in contexts and target densities does not only influence handler expectation, it also influences canine expectations (Gazit, I., Goldblatt, A., & Terkel, J. (2005). The role of context specificity in learning: the effects of training context on explosives detection in dogs. Animal Cognition, 8(3), 143-150.)
